# Adult psychosocial outcomes of men and women who were looked-after or adopted as children: prospective observational study

Alison Teyhan,[1] Dinithi Wijedasa,[2] John Macleod[1]

[1]Population Health Sciences, Bristol Medical School, University of Bristol, Bristol, UK
[2]School for Policy Studies, University of Bristol, Bristol, UK

**Correspondence to**
Dr Alison Teyhan;
alison.teyhan@bristol.ac.uk

## ABSTRACT

**Objective** To investigate whether men and women who were looked-after (in public care) or adopted as children are at increased risk of adverse psychological and social outcomes in adulthood.

**Design, setting** Prospective observational study using the Avon Longitudinal Study of Parents and Children, which recruited pregnant women and their male partners in and around Bristol, UK in the early 1990s.

**Participants** 8775 women and 3654 men who completed questionnaires at recruitment (mean age: women 29; men 32) and 5 years later.

**Exposure** Childhood public care status: looked-after; adopted; not looked-after or adopted (reference group).

**Outcomes** Substance use (alcohol, cannabis, tobacco) prepregnancy and 5 years later; if ever had addiction; anxiety and depression during pregnancy and 5 years later; if ever had mental health problem; social support during pregnancy; criminal conviction.

**Results** For women, 2.7% were adopted and 1.8% had been looked-after; for men, 2.4% and 1.4%, respectively. The looked-after group reported the poorest outcomes overall, but this was not a universal pattern, and there were gender differences. Smoking rates were high for both the looked-after (men 47%, women 58%) and adopted (men 44%, women 40%) groups relative to the reference group (both 28%). The looked-after group were at increased risk of a high depression score (men: 26% vs 11%, OR 2.9 (95% CI 1.5 to 5.6); women: 24% vs 9%, 3.4 (2.2 to 5.0)). A high anxiety score was reported by 10% of the reference women, compared with 26% of those looked-after (3.0 (2.0 to 4.5)) and 17% of those adopted (1.8 (1.2 to 2.6)). Looked-after men and women reported the lowest social support, while criminal convictions and addiction were highest for looked-after men. Adjustment for adult socioeconomic position generally attenuated associations for the looked-after group.

**Conclusions** The needs of those who experience public care as children persist into adulthood. Health and social care providers should recognise this.

## INTRODUCTION

In the UK, the state has a duty to safeguard and promote the welfare of children, to protect them from maltreatment and to promote their health and development.[1] Surveillance

### Strengths and limitations of this study

► The longitudinal, population-based cohort allows comparison of several psychosocial outcomes in adults who experienced being looked-after or adopted as children.

► Several of the outcomes are measured at two timepoints 5 years apart, allowing the persistence or amelioration of disadvantage to be considered.

► We have several measures of adult socioeconomic position, but do not have data on early life factors.

► The cohort only includes parents, which limits generalisability to the wider population of looked-after and adopted adults.

► Adoption and social care practices in England have changed since our participants were children, mainly in the 1960s and 1970s. Therefore, the results may not be generalisable to those who have been adopted or looked-after more recently.

by frontline services aims to identify children in need. Their families can be provided with additional support, but if this is inadequate to mitigate risk or to enable the child's needs to be met, parental responsibility may be taken on by the state. Reasons for this include abuse, neglect, family dysfunction, acute family stress and disability. Children may be looked-after on a short-term or long-term basis, with some experiencing multiple periods of care.[2] Previous research has estimated that the chances of returning home after a year in care are very small; about 8 out of 10 of those who have been looked-after for a year are still looked-after 1 year later.[3]

For most children who need a permanent substitute home in the UK, adoption or long-term fostering becomes the plan. For younger children, adoption is often the preferred long-term care model as it provides a greater level of permanence and a 'family for life'.[4 5] Children who grow up with adopted parents generally report higher levels of emotional

security, sense of belonging and well-being than those in long-term foster care.[6]

For children in long-term care who age out of the care system, the transition to adulthood can be fraught with difficulties. In England, this happens at age 18, with the local authority continuing to provide support and advice until age 21, or 25 if in education or training.[7] Beyond this, these young people often have little financial or social support. Their transition to adulthood is accelerated compared with their peers; they have to live independently, find employment and manage their finances at a younger age, and most often, without the support of a stable family.[8]

Routine statistics and epidemiological studies show that children in care in the UK do worse than their peers across many domains. Notably, they have lower educational attainment,[9–12] poorer mental health[13–16] and are over-represented in the criminal justice system.[17] They are at increased risk of many interlinked adverse circumstances as they enter adulthood, including unemployment, homelessness, social isolation, drug use, self-harm and imprisonment.[8 18 19] These risks may be reduced if children are successfully adopted; however some studies have found that adopted children have more emotional, behavioural and academic problems than their non-adopted peers[20] and experience more bullying in adolescence.[21]

Relatively little research has considered outcomes beyond young adulthood. Two previous studies have used UK cohort data to examine the impact of being looked-after on outcomes in adulthood: one used the 1970 British Birth Cohort Study (BCS70) to examine outcomes in men and women at age 30,[22] the other used the Millennium Cohort Study (MCS) to examine outcomes in women mostly aged 20–39, who were born in the 1960s and 1970s.[23] In both, those who had been in care had increased risk of poorer outcomes in adulthood, for example, lower socioeconomic status, poorer health and more smoking. A further British Cohort Study, the 1958 BCS, has been used to compare outcomes in adulthood between those who were adopted and their peers,[24 25] with the adopted women in particular being found to have positive outcomes. Surprisingly, there have been no UK studies that have compared adult outcomes of children who were adopted out of care versus those in other care placements, despite the fact that both are radical interventions in the lives of children.

Therefore although the poor outcomes of care leavers in the UK are well-recognised, including by the Government,[26] little is known about how such adversities persist or change for this vulnerable group beyond young adulthood. Similarly, although many adopted children do well, there has been little research on how their outcomes in adulthood compare with their looked-after or general population peers. As a consequence, there is a lack of evidence on the additional needs, if any, of care leavers and adoptees at older ages. This paper aims to add to this currently limited evidence base. We use data from another UK population-based cohort study whose adult participants were born mostly in the 1960s. This cohort allowed the examination of outcomes at two time-points in adults who were looked-after and adults who were adopted as children. We chose outcomes which relate to four key areas in which looked-after children or young care leavers are known to experience increased risk: mental health difficulties, substance use, poor social support and criminal conviction.

## METHODS

### Sample

The Avon Longitudinal Study of Parents and Children (ALSPAC) recruited 14541 pregnant women living in a defined area in and around the city of Bristol, UK in 1991–1992. The women have been sent regular postal questionnaires ever since. They were also sent questionnaires for their partner to complete. The women chose who, if anyone, to give these to. For a minority of the women, the partner she gave the questionnaires to changed over time. We therefore only include men in our study who consistently report being the father of the study child to ensure we have data on the same man at all time points. The main results presented in this study are based on 8775 women and 3654 men who returned questionnaires during the pregnancy and when the study child was aged 5 years. More details on ALSPAC are available in the cohort profiles,[27 28] and the searchable data dictionary (http://www.bristol.ac.uk/alspac/researchers/).

### Exposure measures

Experiences of being in care or adopted were reported via questionnaires administered during the pregnancy: adopted when aged<18 years (no, yes); ever been in the care of a local authority or voluntary agency (no; yes; unsure); ever stayed in a children's or residential home (no; <1 week; 1 week–1 month; 1–6 months; >6 months); ever stayed in a foster parents' home (no; yes). Those who responded yes to having been in care, or to having lived in a foster or children's home, were deemed to have been in care. A 3-category exposure variable was derived defining childhood public care status: not looked-after or adopted; looked-after (not adopted); adopted. For the main analyses, the few individuals who reported that they were unsure if they had ever been looked-after (and who did not report that they had lived in children's home or foster care) were included in the looked-after group. A sensitivity analysis was also conducted excluding these individuals.

### Main outcome measures

#### Substance use

Prepregnancy substance use was reported via questionnaires administered during the pregnancy with the study child: smoked regularly (no; yes); drank at least one unit of alcohol per day (no; yes) (1 UK unit is 8 g of alcohol, eg, half a pint of lager)[29]; used cannabis in the 6 months

before pregnancy (no; yes). Participants also reported if they had ever suffered from drug addiction or alcoholism (no; yes). Five years later, participants reported if they: currently smoked; had used cannabis in the past year; currently drank at least one unit of alcohol per day.

## Mental health

During pregnancy, the participants reported if they had ever had schizophrenia, anorexia, severe depression or other psychiatric problem: a variable was derived stating whether the respondent had ever had any of these psychiatric problems (no; yes). Symptoms of depression and anxiety were measured at 18 weeks gestation. Depressive symptoms were measured using the Edinburgh Postnatal Depression Scale. Although this measure was originally designed for use with postnatal women, the 10 items are not specific to women or this period and it has been validated for use at other times.[30] Anxiety symptoms were measured by the 8 items of the anxiety subscale of the Crown-Crisp Experiential Index.[31] Binary variables were derived which determined whether a respondent had a high score (>90% percentile) or not: for women this represented a score ≥14 for depression and ≥10 for anxiety; for men a score ≥10 for depression and ≥7 for anxiety. The women also reported these measures in the same way when their child was aged 5 years. At this time-point, both women and men reported if they had experienced symptoms of anxiety/nerves or depression in the past year.

## Social support and networks

During the pregnancy, the participants reported their social support via their level of agreement to ten statements and their social network via a further 10 statements (details on question wording in online supplementary tables A and B). For both measures, scores were summed with each having a possible range of 0–30. A higher score reflects more social support or a better social network. Binary variables were derived to identify those with poor social support or networks, defined as being in the bottom decile. Cut-offs were the same for men and women: ≤12 for low social support, ≤18 for poor social network.

## Conviction for criminal offence

Participants were asked if they had ever been convicted of a criminal offence in several questionnaires which covered the period from the start of pregnancy, up until the child was aged 5. A binary variable was derived: any conviction (no; yes).

## Other variables

Socioeconomic position (SEP) was measured during the pregnancy. Household occupational social class was based on the lowest class reported by the woman and her partner (I/II (professional/managerial and technical); III (skilled manual and skilled non-manual); IV/V (semiskilled/unskilled manual)).[32] Men and women also reported their highest educational qualification (university degree; A level; O level; vocational/none). Other

measures were reported by women only: financial difficulties (quartiles of score with range 0–40, where 0 is no financial difficulties); housing tenure (owned/mortgaged, private rent, council rent, other); partner status (married; live with partner; do not live with partner; no partner); whether pregnancy with the study child was intentional (no; yes) and their parity (0; 1; 2; 3+). In the 5-year questionnaire, they reported their pregnancy intentions (not pregnant; intend to try later; currently trying; pregnant).

The highest childhood happiness of the men and women was derived from responses to questions on how happy their childhood was at 0–5 years, 6–11 years and 12–15 years (very happy; moderately; quite or very unhappy). School stability was measured by the number of schools attended before the age of 16 years (0–2; 3; 4; 5+). Participants also reported adversities before the age of 17 years: suspended from school; in trouble with police, pregnant (or partner pregnant for the men).

## Analysis

The exposure groups were compared in terms of their adult SEP and childhood experiences using descriptive statistics. Logistic regression models were used to examine associations between the exposure and each of the substance use, mental health, social network and criminal conviction outcomes. Those who had not been adopted or looked-after were the reference group. Models were run unadjusted and adjusted for age and measures related to SEP (relationship status, education, financial difficulties, social class and housing tenure). Models also adjusted for parity for women, and for whether the pregnancy was intentional for the pregnancy time-point outcomes, and for pregnancy intentions for the 5-year time-point outcomes. Where results are described as being 'adjusted' in the results text, this means after adjustment for age, all of the SEP variables, plus parity and pregnancy-intention variables where relevant. All analyses were performed stratified by gender.

A sensitivity, cross-sectional analysis of the measures reported during pregnancy only was performed to determine if results were consistent when the sample was not restricted to those who also participated 5 years later.

## Missing data

Of the 8775 women in the longitudinal sample, 46.7% had complete data and a further 43.8% had between 1 and 4 missing values. Of the 3654 men, 16.5% had complete data, and a further 78.2% had 1–4 missing values. The percentage of missing data is summarised for each variable in online supplementary table C. Multiple imputation using chained equations was used to replace missing data with predictions based on information observed in the sample. All of the variables included in the analyses models, plus other variables associated with missingness or the variables in the model, were included in the imputation models. Imputed datasets (55 for the women, 100 for the men) were created and analysed using

mi estimate commands in Stata V.14.2 (Stata, College Station, Texas, USA).

In the cross-sectional sample for the sensitivity analysis, complete-case analysis was performed. Of the 11 571 women who reported their childhood care status, 7795 had complete outcome data reported during pregnancy, and of those 7088 had complete confounder data. Of the 7676 men who reported childhood care status, 3163 had complete outcome data reported during their partner's pregnancy, of whom 2820 had complete confounder data.

## RESULTS

In the longitudinal sample, 2.4% (95% CI 1.9% to 2.9%) of the men and 2.7% (2.4% to 3.1%) of the women had been adopted. A further 1.4% (1.0% to 1.8%) of the men and 1.8% (1.5% to 2.1%) of the women had been looked-after. These looked-after percentages include the small number of participants who reported that they were unsure if they had been looked-after; excluding them reduced the proportion who were looked-after to 1.1% of men and 1.6% of women.

Of the men who reported their care setting while being looked-after, 39% had lived in foster care only, 44% in a children's home only and 17% in both. Of the women, 30% had lived in foster care, 42% in a children's home and 28% in both. Over 60% of the men and women who had lived in a children's home had done so for >6 months. Of the adoptees, many had been adopted aged <1 year (women 49%, men 63%). A minority of the adoptees had also been in care (women 14%, men 16%), with a similar proportion reporting that they didn't know.

Compared with the reference group, those who had been looked-after were generally younger, less likely to be married and of lower SEP (women table 1; men table 2). With regard to their childhood, they were the least likely to have been very happy and the most likely to have attended multiple schools, to have been suspended and in trouble with the police (online supplementary table D). Women who had been looked-after were the most likely to have been pregnant before the age of 17 and to have had several previous births. They were least likely to report the pregnancy with the study child was intentional, but most likely to be trying to conceive 5 years later. Participants who had been adopted generally had SEP and childhood measures intermediate to those who had been looked-after and those who had not been looked-after or adopted.

| Table 1 | Characteristics of the women by care status in childhood, n=8775 | | | |
|---|---|---|---|---|
| | | **Reference group (95.5%)** | **Looked-after (1.8%)** | **Adopted (2.7%)** |
| Age at delivery | Mean age in years | 28.8 (28.7 to 28.9) | 27.6 (26.6 to 28.5) | 27.7 (27.2 to 28.3) |
| | ≤23 years (%) | 11.9 (11.2 to 12.6) | 26.7 (19.4 to 34.1) | 15.2 (10.3 to 20.2) |
| | ≥34 years (%) | 15.4 (14.6–16.2) | 15.4 (9.3 to 21.4) | 8.7 (5.0 to 12.5) |
| Relationship status | Married (%) | 80.3 (79.4 to 81.2) | 59.8 (51.4 to 68.2) | 78.3 (72.8 to 83.8) |
| | Resident partner (%) | 13.4 (12.7 to 14.2) | 20.4 (13.3 to 27.5) | 12.6 (8.1 to 17.1) |
| | Non-resident partner (%) | 4.4 (4.0 to 4.9) | 14.0 (8.0 to 20.0) | 7.0 (3.5 to 10.5) |
| | No partner (%) | 1.9 (1.6 to 2.2) | 5.8 (1.7 to 9.9) | 2.1 (0 to 4.1) |
| Parity | 0 (%) | 46.0 (45.0 to 47.1) | 36.3 (28.0 to 44.7) | 48.3 (41.7 to 54.9) |
| | 3+ (%) | 4.6 (4.1 to 5.1) | 15.5 (9.3 to 21.6) | 5.3 (2.4 to 8.2) |
| This pregnancy intentional | Yes (%) | 73.3 (72.3 to 74.3) | 53.5 (45.0 to 61.9) | 69.2 (63.1 to 75.4) |
| Pregnancy intentions at 5 years | Pregnant (%) | 3.8 (3.4 to 4.2) | 3.3 (2.5 to 11.3) | 4.8 (1.9 to 7.8) |
| | Trying to get pregnant (%) | 2.2 (1.9 to 2.5) | 6.9 (2.5 to 11.3) | <3 |
| Highest maternal education | Degree (%) | 15.2 (14.4 to 16.0) | <3 | 11.2 (7.0 to 15.4) |
| | Vocational/none (%) | 24.3 (23.3 to 25.2) | 48.8 (39.8 to 57.8) | 21.8 (16.1 to 27.4) |
| Financial difficulties | Q1 (none) (%) | 39.4 (38.3 to 40.4) | 22.8 (15.7 to 29.8) | 29.8 (23.6 to 35.9) |
| | Q4 (high) (%) | 17.7 (16.8 to 18.5) | 33.2 (24.8 to 41.6) | 20.1 (14.7 to 25.5) |
| Housing tenure | Owned/mortgaged (%) | 80.6 (79.8 to 81.5) | 48.1 (39.6 to 56.6) | 74.8 (69.0 to 80.6) |
| Lowest social class of self and partner | I and II (%) | 27.9 (26.9 to 28.9) | 16.7 (10.1 to 23.3) | 25.5 (19.6 to 31.4) |
| | IV and V (%) | 18.1 (17.2 to 18.9) | 26.1 (17.5 to 34.7) | 21.3 (15.6 to 27.1) |

**Table 2** Characteristics of the men by care status in childhood, n=3654

| | | Reference group (96.1%) | Looked-after (1.4%) | Adopted (2.4%) |
|---|---|---|---|---|
| Age when partner at 18 weeks gestation | Mean (years) | 31.5 (31.3 to 31.7) | 30.5 (29.1 to 32.0) | 30.4 (29.1 to 31.8) |
| | ≤23 years (%) | 3.4 (2.8 to 4.1) | <10 | 7.7 (1.7 to 13.6) |
| | ≥34 years (%) | 30.2 (28.6 to 31.7) | 26.8 (13.8 to 39.9) | 23.5 (14.3 to 32.7) |
| Relationship status | Married (%) | 88.1 (87.0 to 89.2) | 78.8 (67.4 to 90.3) | 78.5 (69.7 to 87.2) |
| | Resident partner (%) | 10.3 (9.3 to 11.3) | 21.2 (9.7 to 32.6) | 20.4 (11.8 to 28.9) |
| Highest education | Degree (%) | 29.3 (27.7 to 30.8) | <10 | 18.6 (10.2 to 27.0) |
| | Vocational/none (%) | 19.3 (17.9 to 20.6) | 53.3 (38.8 to 67.8) | 18.3 (9.9 to 26.7) |
| Financial difficulties | Q1 (none) (%) | 47.1 (45.5 to 48.8) | 21.9 (10.1 to 33.7) | 33.5 (23.2 to 43.7) |
| | Q4 (high) (%) | 12.5 (11.4 to 13.7) | 29.2 (16.3 to 42.1) | 19.3 (10.5 to 28.1) |
| Housing tenure | Owned/mortgaged (%) | 87.4 (86.3 to 88.5) | 53.8 (39.8 to 67.9) | 75.0 (65.8 to 84.3) |
| Lowest social class of self and partner | I and II (%) | 35.8 (34.2 to 37.4) | 13.8 (4.0 to 23.7) | 21.7 (12.6 to 30.7) |
| | IV and V (%) | 14.9 (13.7 to 16.1) | 24.9 (12.1 to 37.7) | 17.8 (9.1 to 26.6) |

## Substance use

Substance use results are given in table 3.

Those who had been looked-after or adopted were more likely to be smokers than the reference group at both time-points, with the highest rates observed for looked-after women. Women who had been looked-after or adopted were also more likely to have used cannabis at both time-points. Daily alcohol consumption did not differ between those who had been looked-after and those in the reference group for men or women. Adopted women were the most likely of the women to drink daily prepregnancy, but the adopted men were the least likely at both time-points. Rates of addiction to alcohol or drugs were particularly high in men who had been looked-after or adopted. In women, addiction was also more common in those who had been looked-after or adopted, but numbers were small. Adjustment for age, relationship status and measures of SEP attenuated associations observed for smoking for participants who were looked-after, but had less of an effect on alcohol and cannabis associations or associations for participants who were adopted.

## Mental health

Mental health results are given in table 4.

Women who had been looked-after were the most likely to have high depression and anxiety scores at both time-points, to have ever had a mental health problem and to have experienced symptoms of anxiety and depression in the past year at the later time-point. Women who had been adopted were also more likely to have high anxiety scores and to have had a mental health problem than those in the reference group. Adjustment attenuated results for the looked-after group but not the adopted group. Men who had been looked-after were more likely to have a high depression score during their partner's pregnancy and more likely to have ever had a mental health problem than the reference group. Adjustment for SEP attenuated the differences. At the 5 year time-point, there was no difference by care status in the percentage of men reporting symptoms of anxiety or depression in the previous year. Men in each category were less likely to report these symptoms than women, with the gender difference being particularly large for depressive symptoms.

## Social support and conviction for criminal offence

Social support and criminal involvement results are given in table 5.

Participants who had been looked-after were the most likely to report low social support and a limited social network. Differences relative to the reference group were attenuated on adjustment. Women who had been adopted were more likely to report a poor social network than the reference group, but not a low level of social support. Adopted men did not differ from the reference men.

Men and women who had been looked-after or adopted were more likely to have been convicted of an offence compared with the reference groups. Differences were attenuated on adjustment. Conviction rates were considerably higher for men than women in each of the exposure categories.

## Sensitivity analyses

Analyses were repeated excluding the small number of participants who were unsure if they had been looked-after and results were consistent with those of the main analyses. In cross-sectional, complete-case analyses of the outcomes reported in pregnancy only, a slightly higher proportion of the men had been looked-after (2.4%) or adopted (3.2%) compared with the longitudinal sample. In contrast, proportions for women in the cross-sectional sample (looked-after 1.6%; adopted 2.6%) were similar to

**Table 3** Substance use outcomes by childhood public care status for men and women prepregnancy and when study child aged 5 years

| | | Men | | | Women | | |
|---|---|---|---|---|---|---|---|
| | | | OR (95% CI) | | | OR (95% CI) | |
| | | %† | Unadjusted | Adjusted‡ | %† | Unadjusted | Adjusted§ |
| Has ever had addiction | Reference | 2.0 | Ref | Ref | 1.0 | Ref | Ref |
| | Looked-after | 13.5 | 7.8 (3.4 to 17.8)* | 4.0 (1.6 to 10.0)* | 3.1 | 2.9 (0.9 to 9.1) | 1.6 (0.5 to 5.4) |
| | Adopted | 6.8 | 3.6 (1.5 to 8.6)* | 2.3 (0.9 to 6.0) | 3.0 | 2.9 (1.3 to 6.8)* | 2.8 (1.2 to 6.6)* |
| Prepregnancy | | | | | | | |
| Smoked regularly | Reference | 28.0 | Ref | Ref | 28.3 | Ref | Ref |
| | Looked-after | 46.5 | 2.2 (1.3 to 3.9)* | 1.2 (0.7 to 2.2) | 57.9 | 3.5 (2.5 to 4.9)* | 1.9 (1.3 to 2.8)* |
| | Adopted | 44.3 | 2.0 (1.3 to 3.2)* | 1.7 (1.1 to 2.7)* | 40.3 | 1.7 (1.3 to 2.3)* | 1.6 (1.2 to 2.2)* |
| Drank alcohol daily | Reference | 23.8 | Ref | Ref | 11.3 | Ref | Ref |
| | Looked-after | 21.8 | 0.9 (0.5 to 1.7) | 1.1 (0.6 to 2.3) | 9.6 | 0.8 (0.4 to 1.5) | 0.9 (0.5 to 1.7) |
| | Adopted | 11.4 | 0.4 (0.2 to 0.8)* | 0.4 (0.2 to 0.8)* | 17.0 | 1.6 (1.1 to 2.3)* | 1.8 (1.3 to 2.6)* |
| Used cannabis | Reference | 6.6 | Ref | Ref | 3.9 | Ref | Ref |
| | Looked-after | 9.1 | 1.3 (0.4 to 4.7) | 1.0 (0.3 to 3.6) | 11.2 | 3.1 (1.8 to 5.5)* | 2.3 (1.2 to 4.3)* |
| | Adopted | 9.9 | 1.5 (0.7 to 3.4) | 1.1 (0.5 to 2.7) | 5.5 | 1.4 (0.8 to 2.6) | 1.2 (0.6 to 2.2) |
| When study child 5 years | | | | | | | |
| Smoked regularly | Reference | 22.3 | Ref | Ref | 23.1 | Ref | Ref |
| | Looked-after | 44.2 | 2.8 (1.6 to 4.8)* | 1.6 (0.9 to 2.9) | 52.9 | 3.7 (2.7 to 5.3)* | 2.0 (1.4 to 3.0)* |
| | Adopted | 40.3 | 2.4 (1.5 to 3.6)* | 2.0 (1.2 to 3.2)* | 32.5 | 1.6 (1.2 to 2.1)* | 1.5 (1.1 to 2.1)* |
| Drank alcohol daily | Reference | 35.9 | Ref | Ref | 16.4 | Ref | Ref |
| | Looked-after | 26.9 | 0.7 (0.4 to 1.3) | 1.1 (0.6 to 2.0) | 11.0 | 0.6 (0.4 to 1.1) | 0.9 (0.5 to 1.6) |
| | Adopted | 19.4 | 0.4 (0.3 to 0.8)* | 0.5 (0.3 to 0.9)* | 17.0 | 1.0 (0.7 to 1.5) | 1.2 (0.8 to 1.7) |
| Used cannabis | Reference | 6.1 | Ref | Ref | 4.2 | Ref | Ref |
| | Looked-after | 8.5 | 1.4 (0.5 to 3.9) | 1.0 (0.3 to 2.9) | 8.8 | 2.2 (1.2 to 4.1)* | 1.5 (0.7 to 2.9) |
| | Adopted | 9.9 | 1.7 (0.8 to 3.5) | 1.2 (0.6 to 2.6) | 8.5 | 2.1 (1.3 to 3.5)* | 1.9 (1.1 to 3.1)* |

*P<0.05.

†Percentages rather than n given as these are results from imputed data and so the n differs across the imputed datasets. The percentages shown are from results aggregated across all the imputed datasets using Rubin's rules.

‡Adjusted for age, relationship status, education, financial difficulties, social class, housing tenure.

§Adjusted for age, relationship status, education, financial difficulties, social class, housing tenure, parity, pregnancy intentional (prepregnancy models only), pregnancy status/intentions (5 year models only).

those in the longitudinal sample. The men in the complete case sample were of lower SEP than those in the longitudinal sample (online supplementary table E), but this was not observed for the women (online supplementary table F). Overall, the pattern of results we observed in the longitudinal sample was broadly replicated (online supplementary tables G to I).

## DISCUSSION
### Overview of findings
We used a population-based study to examine substance use, mental health, social support and criminal conviction in adulthood among individuals who had been looked-after or adopted as children, compared with their peers in the sample who were neither adopted nor looked-after. There was a general, but not universal, pattern of these adults reporting more mental health problems, smoking and criminal convictions and less social support than their peers. Overall, the looked-after individuals reported more negative outcomes than those who had been adopted. There were gender differences in some of the associations observed. For example, women who had been looked after had high rates of anxiety, whereas men who had been looked-after had an excess risk of addiction and criminality. Adjustment for SEP measures often

**Table 4** Mental health outcomes by childhood public care status for men and women during pregnancy and when study child aged 5 years

| | | Men | | | Women | | |
|---|---|---|---|---|---|---|---|
| | | | OR (95% CI) | | | OR (95% CI) | |
| | | %† | Unadjusted | Adjusted‡ | %† | Unadjusted | Adjusted§ |
| Has ever had mental health problem | Reference | 5.7 | Ref | Ref | 10.2 | Ref | Ref |
| | Looked-after | 24.1 | 5.3 (2.7 to 10.2)* | 3.9 (1.9 to 7.8)* | 25.9 | 3.1 (2.1 to 4.5)* | 2.2 (1.4 to 3.2)* |
| | Adopted | 12.4 | 2.4 (1.2 to 4.5)* | 2.0 (1.0 to 3.9) | 15.5 | 1.6 (1.1 to 2.3)* | 1.6 (1.1 to 2.3)* |
| During pregnancy | | | | | | | |
| High anxiety score | Reference | 9.3 | Ref | Ref | 10.4 | Ref | Ref |
| | Looked-after | 15.2 | 1.7 (0.8 to 3.9) | 1.6 (0.7 to 3.7) | 26.1 | 3.0 (2.0 to 4.5)* | 2.0 (1.3 to 3.1)* |
| | Adopted | 11.1 | 1.2 (0.6 to 2.4) | 1.2 (0.6 to 2.4) | 17.4 | 1.8 (1.2 to 2.6)* | 1.6 (1.1 to 2.4)* |
| High depression score | Reference | 11.0 | Ref | Ref | 8.7 | Ref | Ref |
| | Looked-after | 26.2 | 2.9 (1.5 to 5.6)* | 2.3 (1.1 to 4.5)* | 24.3 | 3.4 (2.2 to 5.0)* | 2.0 (1.3 to 3.1)* |
| | Adopted | 12.3 | 1.1 (0.6 to 2.2) | 1.0 (0.5 to 1.9) | 10.9 | 1.3 (0.8 to 2.0) | 1.1 (0.7 to 1.8) |
| When study child 5 years | | | | | | | |
| High anxiety score | Reference | / | / | / | 10.1 | Ref | Ref |
| | Looked-after | / | / | / | 18.7 | 2.0 (1.3 to 3.2)* | 1.5 (1.0 to 2.5) |
| | Adopted | / | / | / | 14.3 | 1.5 (1.0 to 2.2)* | 1.4 (1.0 to 2.1) |
| High depression score | Reference | / | / | / | 9.5 | Ref | Ref |
| | Looked-after | / | / | / | 18.5 | 2.2 (1.4 to 3.4)* | 1.6 (1.0 to 2.5)* |
| | Adopted | / | / | / | 12.2 | 1.3 (0.9 to 2.0) | 1.3 (0.8 to 1.9) |
| Anxiety symptoms in past year | Reference | 18.4 | Ref | Ref | 22.5 | Ref | Ref |
| | Looked-after | 17.1 | 0.9 (0.4 to 1.9) | 1.0 (0.5 to 2.2) | 31.4 | 1.6 (1.1 to 2.3)* | 1.4 (1.0 to 2.1) |
| | Adopted | 17.8 | 1.0 (0.5 to 1.7) | 1.0 (0.5 to 1.7) | 25.8 | 1.2 (0.9 to 1.6) | 1.2 (0.9 to 1.6) |
| Depression symptoms in past year | Reference | 12.6 | Ref | Ref | 23.0 | Ref | Ref |
| | Looked-after | 15.0 | 1.2 (0.6 to 2.7) | 1.0 (0.4 to 2.2) | 38.8 | 2.1 (1.5 to 3.0)* | 1.6 (1.1 to 2.3)* |
| | Adopted | 8.2 | 0.6 (0.3 to 1.4) | 0.5 (0.2 to 1.2) | 27.3 | 1.3 (0.9 to 1.7) | 1.2 (0.9 to 1.6) |

*P<0.05.

†Percentages rather than n given as these are results from imputed data and so the n differs across the imputed datasets. The percentages shown are from results aggregated across all the imputed datasets using Rubin's rules.

‡Adjusted for age, relationship status, education, financial difficulties, social class, housing tenure.

§Adjusted for age, relationship status, education, financial difficulties, social class, housing tenure, parity, pregnancy intentional (prepregnancy models only), pregnancy status/intentions (5 year models only).

attenuated associations substantially for the looked-after group, but generally had less of an impact for the adopted group. Rather than implying that adult SEP confounds the relationship between childhood care status and adult psychosocial outcomes, this suggests that it in part mediates these effects.

### Comparison with previous literature—adults who have been looked-after

In the UK, the poor outcomes for young care leavers have been widely reported, both in academic literature and increasingly in mainstream media.[33–37] Our study is one of the few to consider outcomes at an older age. The high rate of depressive symptoms in adults who had been looked after as children in our sample was in concordance with both the BCS70[22] and MCS[23] studies. Neither included measures of anxiety with which to corroborate our finding of high anxiety in looked-after women but not men.

While the looked-after men reported more addiction in our sample, it was for the looked-after women that we found the strongest association with cannabis use. In

**Table 5** Social support and criminal involvement outcomes by childhood public care status for men and women during pregnancy and up to when study child aged 5 years

| | | Men | | | Women | | |
|---|---|---|---|---|---|---|---|
| | | | OR (95% CI) | | | OR (95% CI) | |
| | | %† | Unadjusted | Adjusted‡ | %† | Unadjusted | Adjusted§ |
| During pregnancy | | | | | | | |
| Low social support | Reference | 10.8 | Ref | Ref | 8.1 | Ref | Ref |
| | Looked-after | 25.2 | 2.8 (1.4 to 5.4)* | 2.2 (1.1 to 4.4)* | 22.5 | 3.3 (2.1 to 5.0)* | 1.6 (1.0 to 2.5) |
| | Adopted | 8.7 | 0.8 (0.4 to 1.7) | 0.7 (0.3 to 1.5) | 10.5 | 1.3 (0.8 to 2.1) | 1.2 (0.7 to 2.0) |
| Poor social network | Reference | 12.7 | Ref | Ref | 8.8 | Ref | Ref |
| | Looked-after | 26.4 | 2.5 (1.3 to 4.9)* | 1.9 (1.0 to 3.8) | 26.3 | 3.7 (2.5 to 5.5)* | 2.0 (1.3 to 3.1)* |
| | Adopted | 14.1 | 1.2 (0.6 to 2.1) | 1.1 (0.6 to 2.1) | 13.2 | 1.6 (1.0 to 2.3)* | 1.5 (1.0 to 2.3) |
| From pregnancy to child aged 5 | | | | | | | |
| Any criminal conviction | Reference | 6.9 | Ref | Ref | 1.4 | Ref | Ref |
| | Looked-after | 23.1 | 3.9 (1.5 to 10.4)* | 2.6 (0.9 to 7.5) | 6.2 | 4.4 (1.7 to 11.5)* | 2.4 (0.9 to 6.5) |
| | Adopted | 15.0 | 2.3 (1.1 to 5.0)* | 1.8 (0.8 to 4.2) | 4.6 | 3.2 (1.4 to 7.5)* | 2.9 (1.2 to 7.1)* |

*P<0.05.

†Percentages rather than n given as these are results from imputed data and so the n differs across the imputed datasets. The percentages shown are from results aggregated across all the imputed datasets using Rubin's rules.

‡Adjusted for age, relationship status, education, financial difficulties, social class, housing tenure.

§Adjusted for age, relationship status, education, financial difficulties, social class, housing tenure, parity, pregnancy intentional (prepregnancy models only), pregnancy status/intentions (5 year models only).

a qualitative study of care leavers in England, cannabis was viewed as 'relatively harmless', 'acceptable' and 'to have little impact on parenting': some women said it helped them cope with the stress of looking after a baby.[8] This could explain our finding of a similar prevalence of cannabis use prepregnancy and postpregnancy, in contrast to the decline observed for smoking. Less than 10% of the looked-after individuals in ALSPAC reported cannabis use, a considerably lower percentage than observed in the young care leavers and in agreement with their general finding that substance use declines with age. The numbers reporting use of other illegal substances was too small to analyse, however the 'ever had an addiction' measure is likely to include addictions to other illegal substances. The BCS70 study included a measure of 'illegal substance use in past year', without further breakdown, and found high levels in the men compared with ALSPAC; 26% of those never in care and 34% of those who had been in care.[22] Part of the discrepancy in drug use prevalence between the two studies could reflect differences in sample selection: the ALSPAC sample were recruited to the study as adults who were expecting a baby, whereas the BCS70 sample were in that study from birth, as it was their parents who were recruited.

Smoking rates were very high for looked-after men and women in ALSPAC, consistent with previous studies: of the looked-after women in the MCS, 73% had ever been smokers and 58% smoked during pregnancy;[23] in the care leavers study, two-thirds were daily smokers.[8] In contrast,

we did not find daily alcohol consumption to be associated with looked-after status in ALSPAC. Our measure does not capture binge drinking behaviours, and reverse causation could be an issue whereby those with alcohol problems had stopped drinking. However, our results are consistent with the BCS70 study, where adults who had been looked-after were no more likely to have problems with alcohol.[22]

The looked-after men and women in ALSPAC were the most likely to report a poor social network and limited social support. The instability of life in care can make it difficult to build and maintain friendships and a support network, and young people can reach adulthood with no family and no social base.[6] This reduced social capital exacerbates their difficulties in transitioning to a successful, independent adult life. Our results show that reduced social capital can persist for many years.

The majority of looked-after children in the UK do not receive a criminal conviction, but as a group they are over-represented in the criminal justice system.[17] In our sample, almost a quarter of the looked-after men had been convicted of an offence in the 5-year period considered. Rates for looked-after women, although lower than the men, were also elevated. Our results highlight that for men in particular, criminal involvement continues for a substantial minority of those who were looked-after, even once they have started a family of their own. Findings were consistent in the BCS70.[22]

## Comparison with previous literature—adults who have been adopted

Comparing outcomes in those adopted versus those who remain in the care system, and identifying causal processes, is complicated because those who successfully complete the adoption process may be systematically different from those who remain in long-term care, and these differences may influence their outcomes in addition to any independent influence of the model of care they receive. The adult SEP of the adoptees in ALSPAC was higher than the looked-after group but lower than the reference group. In contrast, adopted women in the 1958 British birth cohort had a higher SEP than the general population at age 33, but this was not true for the men.[24]

For substance use, no clear pattern emerged when comparing those who had been adopted with those who had been in care. Smoking prevalence and cannabis use were similar for the men, and addiction was similar for the women. Of all the groups, daily alcohol consumption was consistently lowest in the adopted men, whereas of the women those who had been adopted were the most frequent drinkers prepregnancy. For men and women in the British 1958 cohort, alcohol problems did not differ between adoptees and the general population.[24] Another study used the 1958 cohort data and similarly found no difference in alcohol abstinence, but the adoptees were more likely to have smoked than the general population. This study also included an additional comparison group, adults who had been adopted from Hong Kong orphanages by British parents; this group were the least likely to drink alcohol or smoke.[25]

Adoptees in ALSPAC were more likely to report having ever had a mental health problem than the reference group, but less likely than the looked-after group; in contrast the adopted men and women in the 1958 birth cohort did not report excess past emotional problems.[24] Adopted women in ALSPAC had high anxiety scores at both time-points relative to the reference group, but again this was not as high as that of the looked-after women. The adopted men did not have higher anxiety scores than the reference men, and neither gender had excess risk of depression. In the 1958 cohort, the adoptees did not have a higher risk of current depressive symptoms[24] or other mental health problems.[25]

Adopted men and women in our study were not more likely than the reference group to have low social support, but the adopted women were more likely to have a limited social network although not to the extent observed for the looked-after women. In the 1958 British cohort, there was a gender difference: adopted women had the highest level of social support, but adopted men the lowest.[24]

The preference for adoption versus long-term foster care differs between countries,[38 39] and there are many other differences internationally in child protection systems, and cultural and social norms regarding out-of-home care and adoption. An added complication when considering adult outcomes is that the exposure happened many years before, and care and adoption

systems have changed over time within and between countries. For these reasons, we have focused our paper on the UK context. However adult outcomes of those who have been in care or adopted have been studied in other countries, including Sweden[40–42] and the USA.[43]

## Role of early childhood adversity

Many looked-after children have experienced adverse childhood experiences (ACEs), which are associated with poor long-term outcomes irrespective of whether the children experiencing these adversities are subsequently looked-after.[44] For many children being looked-after or adopted is likely to be beneficial. For example, educational attainment may improve after long-term care.[10] The generally worse outcomes we have found in adults who have been in care should not be interpreted as strong evidence of adverse effects of care itself. Rather, having been in care is a marker of substantial early childhood disadvantage. A recent paper which examined ACEs in the ALSPAC women, without considering childhood care status, found higher levels of mental illness and smoking and poorer education and social support, in those who had experienced maltreatment in childhood.[45] The results mirror ours and provide evidence for ACEs being important factors in the associations we observe. We were unable to directly examine the role of ACEs in our study as it is not possible to determine if the measures relate to a time before or after a child entered the care system. It is important to acknowledge that becoming looked-after is not inevitably positive for children. Some aspects of care, for example experiences of abuse or neglect within the care system, may compound the effects of early childhood adversity.[46]

## Strengths and limitations

Strengths of our study include its basis in the general population and our ability to examine outcomes at two points in adulthood in participants who reported being looked after compared with those who reported being adopted. Including outcomes at two points in time allowed examination of stability and change in the outcomes we considered. Our study also has limitations that we acknowledge. ALSPAC does not include adults who are not parents. To be eligible, women needed to be pregnant, and men needed to be invited by their pregnant partner to participate. Adults with more problematic lives may be under-represented in our sample. In women, this could reflect lower engagement with antenatal services, through which ALSPAC recruited its sample. In men, it is possible that those who had a difficult or unstable relationship with their child's mother will be less likely to be in the study. These considerations lead us to expect that the associations we found will be underestimates of those in the wider looked-after community. The participants were children mostly in the 1960s and 1970s, and care procedures have changed over time, notably in the greater use of foster as opposed to group care[4] and extended support until age 25.[47] Furthermore, whereas

in the past many adoptions were of relinquished babies, now most are of children aged 1–4 years who are in care due to neglect or abuse.[48] Therefore, our results may not be generalisable to those currently in care or adopted in the last three decades. The ALSPAC measures on looked-after status and adoption lack detail. The age a child enters care, and the number of placements they have, are thought to be key factors in determining the likelihood of positive outcomes,[49] but we do not have this information on our participants. We do not know the reason for their care status or the age they left the care system. Some of the adoptions may have been by step-parents, rather than as a result of being removed from birth parents. Some of the adopted individuals had also been in care, but we do not know why or for how long. As only a small percentage of children experience care or adoption, numbers in a population-based cohort, even one as large as ALSPAC, will be small.

## Implications for practice and policy

Our results are likely of most use to health and social care providers, and evidence a need for support mechanisms to be in place for care leavers beyond young adulthood. Our data do not allow us to made recommendations beyond this. As discussed, we have limited information on the type of care received, at what age, or how long for. We therefore cannot make policy recommendations with regard to these factors. But what we can conclude is that those who experience the care system continue to have poorer outcomes in adulthood than their peers, many years after they have left the care system and to when they have children of their own. We have also shown that adopted individuals have excess risk in some areas, including smoking, addiction and mental health.

Long-term, continued improvements to the care system are needed to maximise the life chances of future children who experience adversity in childhood and are not able to grow up with their birth parents. In particular, evidence is needed on which modifiable aspects of care, beyond permanence and placement stability, promote better outcomes. In the meantime, it must be recognised that some of today's adults who experienced the care system as children have higher needs than those in the general population. In particular, our results suggest a particular need for mental healthcare, social and educational support and for services to reduce the harms associated with substance use.

By highlighting the currently limited evidence base in this area, and the limitations of the ALSPAC data, we hope that our work will also encourage other researchers with suitable data to consider undertaking similar analyses. A stronger evidence base would ultimately allow more specific policy recommendations to be made with the ultimate goal of improving the long-term life chances of those unable to grow up with their birth families.

**Acknowledgements** We are grateful to all the families who took part in ALSPAC, the midwives for their help in recruiting them and the ALSPAC team, which includes interviewers, computer and laboratory scientists, clerical workers, research scientists, volunteers, managers, receptionists and nurses.

**Contributors** AT conceived the study, conducted the analyses, interpreted the data and drafted the manuscript. JM helped develop the research question, interpreted the data and critically revised the paper. DW interpreted the data and critically revised the paper. All authors have read and approved this final version. AT will serve as guarantor for the contents of this paper.

**Funding** The UK Medical Research Council and the Wellcome Trust (Grant Ref: 092731) and the University of Bristol provide core support for ALSPAC. AT and JM are supported by PEARL (Project to Enhance ALSPAC through Record Linkage), a programme of research funded by the Wellcome Trust (WT086118/Z/08/Z), and The Farr Institute CIPHER, which is supported by a 10-funder consortium: Arthritis Research UK, the British Heart Foundation, Cancer Research UK, the Economic and Social Research Council, the Engineering and Physical Sciences Research Council, the Medical Research Council, the National Institute of Health Research, the National Institute for Social Care and Health Research (Welsh Assembly Government), the Chief Scientist Office (Scottish Government Health Directorates), and the Wellcome Trust (MRC Grant No: MR/K006525/1).

**Competing interests** JM is a foster carer. AT and DW have no competing interests.

**Patient consent** Not required.

**Ethics approval** Ethical approval for ALSPAC was obtained from the ALSPAC Ethics and Law Committee and the Local Research Ethics Committees (LREC). Full LREC details are available online (http://www.bristol.ac.uk/alspac/researchers/research-ethics/). This study was approved by the ALSPAC Executive Committee. It involves secondary analysis of ALSPAC questionnaire data, which is fully anonymised before researchers can access or analyse it. No clinical or administrative records were used in this study.

**Provenance and peer review** Not commissioned; externally peer reviewed.

**Data sharing statement** The ALSPAC data management plan (available here: http://www.bristol.ac.uk/alspac/researchers/data-access/ documents/alspac-data-management-plan.pdf) describes in detail the policy regarding data sharing, which is through a system of managed open access. Requests for the data used in this paper can be made to the Executive (alspac-exec@bristol.ac.uk).

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
