## [Reviewer comments · BMJ Open]

ARTICLE DETAILS

TITLE (PROVISIONAL)	Adult psychosocial outcomes of men and women who were looked-after or adopted as children: prospective observational study
AUTHORS	Teyhan, Alison; Wijedasa, Dinithi; Macleod, John

VERSION 1 – REVIEW

REVIEWER	Dr Lisa Moran Institute for Lifecourse and Society School of Political Science and Sociology NUI Galway and Department of Social Sciences Edge Hill University, Ormskirk, Lancashire, UK
REVIEW RETURNED	23-Aug-2017

GENERAL COMMENTS	This is a very interesting paper and the methodology is somewhat novel. It makes some interesting and important points about children in care and/or who are adopted, with regards to their outcomes. All this is encouraging. However, I advise the authors to take account for the following when they are resubmitting their article/reformulating it; 1. Most of the points in the early part of the paper about looked after children (predominantly) having poorer outcomes is well-known in the literature. This part should be reframed to 'sell' the contribution of the paper to this literature, and in particular, the significance of the scales and the methodology.2. The authors should also remember that many children in care internationally have better (or similar) outcomes as their peers who were not in care. There are many issues which affect this, including the age that the child goes into care at, the quality of the care placement, and placement matching which are often mentioned in the literature, and which have spawned a plethora of work since the 1980s at least. Reference to this literature should be made, in my view.3. The methodology is a key selling point of the paper but it is not described in enough depth to 'sell' it. It is also unclear as to what the initial rationale was for the study, whether there was something significant about it being done in Bristol, and what the key focus of the paper is, overall.
---

	4. There are interesting findings about pre-pregnancy substance abuse, mental health and depressive symptoms. I think the paper would benefit from a more comprehensive interaction (or at least a fuller discussion) of how these findings fit with any comparable international findings (if any) and what they might mean for the health status of children and young people who are in care, and their long-term outcomes. 5. There is insufficient information on the measurement scales in my view, which is disappointing. This would give the paper greater clarity and focus. 6. There are some small grammatical errors throughout but these are fixable. 7. There are some questions about the robustness of the methodology, given that the questionnaires for males were given to them by their spouses or partners. There is also limitations in the extent that only men who identified themselves as the child's father at both time points are included. This is acknowledged in the limitations later on but not in any depth. I do acknowledge that the authors are under word constraints however. 8. There is some discussion of the findings in relation to key literature in a subsequent section. This could go deeper into the findings and the significance of the paper, I think. 9. There are some useful points made about permanence in the final section which are factually correct and interesting, and some good observations on the care system. I think that this would benefit as well from some more mention of the significance of this study per se and of future studies that use the same/similar methodologies into the future. 10. This could make an excellent contribution to the literature but in my view, the current draft needs some work.
--	--

REVIEWER	Jessica A.K. Matthews University of Massachusetts, Amherst United States
REVIEW RETURNED	19-Oct-2017

GENERAL COMMENTS	The study has a number of positives: the longitudinal nature, the multiple comparison groups, and the larger sample size. You have a very straightforward project here, and I think it could yield a lot of rich information and contribute a great deal to the adoption field. There are few adoption studies that are longitudinal and have a comparison community sample. I have some notes and comments: On page five, "consistently reported being the father" - what does this mean? How is this defined? I think this needs to be explained/described much more clearly. Obviously nothing can be done at this point in time, but why only ask for length of stay in residential homes and not request information on age at adoption, time in foster care, etc? This should be mentioned in the limitations and addressed in the discussion. You may see differences in results for children adopted at earlier ages, adopted from care, who spent longer in care, etc.
---

	Additionally, another comment that is not possible to address, but your substance use questions seem basic - only address cigarette smoking, alcohol and cannabis - what about other illegal drugs? How did the study deal with children who were adopted from care? Were they grouped with the adoptees or the looked-after group? This needs to be addressed in description of methods/participants. It should also be addressed in discussion as a potential confound/overlap in sample. A few times in the paper, "adjustment" is mentioned. I would assume the authors mean "adjustment for SEP" but this is unclear. This issue arises on page 9 three times. The reporting of your results is very basic, and the tables lack statistical information. Where are the logistic regression coefficients? Standard errors? Wald statistics and degrees of freedom? I see this as a crucial omission and would not accept any paper without these data. In the discussion section - does the BCS70 report on only cannabis use or does it include other illegal drugs - this needs to be more clearly specified for readers who are not intimately familiar with BCS70. Additionally, the timing of the BCS70 needs to be addressed with respect to your comparisons. Is it really that there is low drug use in your sample, or are there chronosystem differences in population drug use? The discussion section needs much more analysis, interpretation and conclusion. This and the statistical omissions are the greatest issues with this manuscript. What do your results mean, and what should we do with this information? On page 11, the paragraph on mental health should likely include a comment about adoptive family follow-up. Research suggests that adoptees and adoptive families may be more comfortable accessing services due to the intimate service use during and immediately following the adoption process. It could be that adopted persons are more comfortable using mental health services that would prevent any mental health issue from escalating to your high thresholds. Your tables could use better delineation of the sample sizes in cells/rows. Finally, given that you have a large sample size and binary indicators, it would have been very interesting to see a latent class analysis with these data. The logistic regression is basic and your sample could lend itself to much more complicated and rich analyses.
--	--

VERSION 1 – AUTHOR RESPONSE

Reviewer: 1, Dr Lisa Moran

This is a very interesting paper and the methodology is somewhat novel. It makes some interesting and important points about children in care and/or who are adopted, with regards to their outcomes.

All this is encouraging. However, I advise the authors to take account for the following when they are resubmitting their article/reformulating it;

1. Most of the points in the early part of the paper about looked after children (predominantly) having poorer outcomes is well-known in the literature. This part should be reframed to 'sell' the contribution of the paper to this literature, and in particular, the significance of the scales and the methodology.

Response: Having reconsidered our Introduction, we agree with the reviewer that it was too focused on the well-established poorer outcomes in early life and did not adequately highlight that the main original contribution of our paper was the focus on outcomes in later adulthood. We have now redrafted the Introduction, with the main changes being the shortening of the parts which describe the care system and early-life poor outcomes, and the inclusion of two new paragraphs – one summarising the limited research to date in this area (which previously was not introduced until the Discussion), and the other explaining gaps in current knowledge, the aims of this study in relation to them, and why we chose to use this cohort study.

2. The authors should also remember that many children in care internationally have better (or similar) outcomes as their peers who were not in care. There are many issues which affect this, including the age that the child goes into care at, the quality of the care placement, and placement matching which are often mentioned in the literature, and which have spawned a plethora of work since the 1980s at least. Reference to this literature should be made, in my view.

Response: The reviewer raises a valid point regarding international differences in child safe-guarding and care systems. There are many relevant variables which differ markedly between countries (e.g. cultural and social norms/attitudes towards being raised in out of home care/adopted; the ways in which child protection/safe-guarding procedures operate; the preference for adoption versus long-term foster or institutional care; the reasons for a child not being raised with their birth family; and the age that Government support ceases). After consideration, we have decided it is beyond the scope of this paper to compare and contrast care systems and their outcomes internationally. To remove any ambiguity, we have now changed the introduction to only summarise UK-based studies (or reviews which include UK studies). Paragraph 3 now begins:

“Routine statistics and epidemiological studies show that children in care in the UK do worse than their peers across many domains.”

Similarly, in the Discussion, we have removed the Swedish studies from the sections which compare our results with previous studies, and instead include the UK literature only. We have added the following paragraph to the end of the adoption comparison section to highlight this issue:

“The preference for adoption versus long term foster care differs between countries (REFS), and there are many other differences internationally in child protection systems and cultural and social norms regarding out-of-home care and adoption. An added complication when considering adult outcomes is that the exposure has happened many years before, and how care and adoption systems have changed over time will also differ between countries.

For these reasons, we have focused our paper on the UK context. However adult outcomes of those who have been in care or adopted have been studied in other countries, including Sweden (REFS) and the US (REF).”

3. The methodology is a key selling point of the paper but it is not described in enough depth to 'sell' it. It is also unclear as to what the initial rationale was for the study, whether there was something significant about it being done in Bristol, and what the key focus of the paper is, overall.

Response: Thank you for raising this important point regarding making the rationale clear. We have now re-written the final paragraph of the Introduction:

“Therefore although the poor outcomes of care leavers in the UK are well-recognised, including by the Government (REF), little is known about how such adversities persist or change for this vulnerable group beyond young adulthood. Similarly, although many adopted children do well, there has been little research on how their outcomes in adulthood compare to their looked-after or general population peers. As a consequence, there is a lack of evidence on the additional needs, if any, of care leavers and adoptees at older ages.”

Later in this paragraph, we also highlight that we chose to use this particular cohort study as it allowed the inclusion of adults who were adopted and those who were looked-after (in contrast to the previous studies which are now summarised in the previous paragraph of the Introduction) and the examination of a range of psychosocial outcomes at two time points. Therefore the choice to use ALSPAC was due to its strengths as a cohort study as opposed to any special significance with regards to the city of Bristol. The ALSPAC study aimed to have a sample representative of the whole of Great Britain, so that results from studies using the data would be generalisable beyond the Bristol area. A number of studies have been undertaken to assess the representative nature of the ALSPAC sample. They have shown that the population of parents and children living in the study area in 1970 were broadly similar to those of the rest of Great Britain. In terms of those recruited to the study, similar to all studies where a representative sample has been attempted, ALSPAC had a slight shortfall in the recruitment of less affluent families and of ethnic minority mothers. More details on sample representativeness available here: <http://www.bristol.ac.uk/alspac/researchers/cohort-profile/>

With regards the depth of our methodology description, our Methods section is already long (around 1300 words), and we felt sufficiently detailed. An overview of the ALSPAC sample is given in the first paragraph of the Methods in the format recommended by the ALSPAC Executive, and references given to the cohort profiles and website which provide more in-depth information. All of the measures (outcome, exposure, and confounder) are described, our approach to missing data detailed, and our analysis approach given.

4. There are interesting findings about pre-pregnancy substance abuse, mental health and depressive symptoms. I think the paper would benefit from a more comprehensive interaction (or at least a fuller discussion) of how these findings fit with any comparable international findings (if any) and what they might mean for the health status of children and young people who are in care, and their long-term outcomes.

Response: The focus of this paper is on understanding the impact of the UK care system on today's adults rather than predicting outcomes of children currently in care, which is a very important issue but not one these particular data are able to address. This is because of changes over time in both the characteristics of children in the care system (e.g. in terms of their need, age at entering care) and in the care system itself (e.g. now a greater use of foster care rather than group care). Similarly, as also mentioned in the discussion, adoption practices have changed over time with children now being more likely to be adopted from care and at an older age than previously.

Consequently, our focus is on today's adults who were adopted, rather than current children who are living with adoptive parents. We have amended a sentence in the section of the discussion which covers these points to add further clarification:

“...Therefore our results may not be generalisable to those currently in care, or adopted in the last three decades.”

Response: Similarly, we do not have appropriate data to examine the potential causes of adverse outcomes in adulthood. We acknowledge the reviewers point that early life and maternal factors such as substance abuse, mental health and depressive symptoms likely have an important role, and we discuss our lack of early life data in our section in the Discussion on 'role of early childhood adversity'. However, we feel that it is not appropriate to have a section discussing these maternal factors in our discussion as it is beyond the scope of our paper, which is already long.

5. There is insufficient information on the measurement scales in my view, which is disappointing. This would give the paper greater clarity and focus.

Response: The Edinburgh Postnatal Depression Scale and Crown-Crisp Experiential Index are well-established, validated, measurement scales and we have provided a reference for both in the Mental Health outcome variables section of the Methods.

The 10 items which comprise the social support and social network scales have now been detailed in an additional file (supplementary Tables A and B). We have removed details of the question wording from the text and replaced with a reference to this supplementary file to avoid duplication:

“During the pregnancy, the participants reported their social support via their level of agreement to ten statements, and their social network via a further ten statements (details on question wording in Supplementary File 2).”

There is a searchable data dictionary available online where researchers can see details on all ALSPAC measures, including question wording and frequencies/means. The webpage where this can be accessed is described in the Sample section of the methods:

“More details on ALSPAC are available in the cohort profiles (REFS), and the searchable data dictionary (www.bristol.ac.uk/alspac/researchers/).”

6. There are some small grammatical errors throughout but these are fixable.

Response: We have proof-read our final document, and believe it is now grammatically correct.

7. There are some questions about the robustness of the methodology, given that the questionnaires for males were given to them by their spouses or partners. There is also limitations in the extent that only men who identified themselves as the child's father at both time points are included. This is acknowledged in the limitations later on but not in any depth. I do acknowledge that the authors are under word constraints however.

Response: We agree with the reviewer that the ALSPAC selection criteria means that the men and women who were included in our sample may not be representative of the wider looked after or adopted population. The selection criteria are a result of ALSPAC being a birth cohort study, thereby only including parents.

We have tried to be as explicit about this limitation as possible. With regards the limitation of us only being able to include men who identified as the study child's father, this point was also raised by Reviewer 2 below. We have added extra detail in the methods section which we hope makes it clear why this was necessary:

“The women have been sent regular postal questionnaires ever since. They were also sent questionnaires for their partner to complete. The women chose who, if anyone, to give these to. For a minority of the women, the partner she gave the questionnaires to changed over time. We therefore

only include men in our study who consistently report being the father of the study child to ensure we have data on the same man at all time points.”

We have also extended the discussion of the impact of the selection criteria in the Limitations section of the Discussion:

“Adults with more problematic lives may be underrepresented in our sample. In women this could reflect lower engagement with antenatal services, through which ALSPAC recruited its sample. In men it is possible that those who had a difficult or unstable relationship with their child’s mother will be less likely to be in the study. These considerations lead us to expect that the associations we found will be under-estimates of those in the wider looked-after community.”

8. There is some discussion of the findings in relation to key literature in a subsequent section. This could go deeper into the findings and the significance of the paper, I think.

Response: We now better highlight the significance of our study in the ‘Implications for practice and policy’ section of the discussion, which we have substantially revised:

“Our results are likely of most use to health and social care providers, and evidence a need for support mechanisms to be in place for care leavers beyond young adulthood. Our data do not allow us to make recommendations beyond this, and we believe it is important to not over-interpret our findings in light of the data limitations. As discussed, we have limited information on the type of care received, at what age, or how long for. We therefore cannot make policy recommendations with regards to these factors. But what we can conclude is that those who experience the care system continue to have poorer outcomes in adulthood than their peers, many years after they have left the care system, and to when they have children of their own. We have also shown that adopted individuals have excess risk in some areas.”

9. There are some useful points made about permanence in the final section which are factually correct and interesting, and some good observations on the care system. I think that this would benefit as well from some more mention of the significance of this study per se and of future studies that use the same/similar methodologies into the future.

Response: As mentioned above, we have now revised our ‘Implications for practice and policy’ section. This also now includes a paragraph recommending future research:

“By highlighting the currently limited evidence base in this area, and the limitations of the ALSPAC data, we hope that our work will also encourage other researchers with suitable data to consider undertaking similar analyses. A stronger evidence base would ultimately allow more specific policy recommendations to be made with the ultimate goal of improving the long term life chances of those unable to grow up with their birth families.”

10. This could make an excellent contribution to the literature but in my view, the current draft needs some work.

Response: Thank you.

Reviewer: 2, Jessica A.K. Matthews

The study has a number of positives: the longitudinal nature, the multiple comparison groups, and the larger sample size. You have a very straightforward project here, and I think it could yield a lot of rich

information and contribute a great deal to the adoption field. There are few adoption studies that are longitudinal and have a comparison community sample.

I have some notes and comments:

1. On page five, "consistently reported being the father" - what does this mean? How is this defined? I think this needs to be explained/described much more clearly.

Response: When ALSPAC began, it did not directly recruit the pregnancy women's partners. Instead, questionnaires for the partner were posted to each woman for her to pass on. She chose who, if anyone, to give them to. That person could therefore change over time. We have re-worded some of the sentences in the sample description section, and added some detail, to clarify this issue for readers:

"... They were also sent questionnaires for their partner to complete. The women chose who, if anyone, to give these to. For a minority of the women, the partner she gave the questionnaires to changed over time. We therefore only include men in our study who consistently report being the father of the study child to ensure we have data on the same man at all time points."

2. Obviously nothing can be done at this point in time, but why only ask for length of stay in residential homes and not request information on age at adoption, time in foster care, etc? This should be mentioned in the limitations and addressed in the discussion. You may see differences in results for children adopted at earlier ages, adopted from care, who spent longer in care, etc.

Response: ALSPAC was not designed to specifically study experiences of being in care or adopted; it is a hypothesis-free, longitudinal study that has collected data on a huge range of topics over the past 26 years. Consequently, the care and adoption measures lack the level of detail that we would have liked. The reviewer is correct that unfortunately nothing can be done to rectify this now. The lack of detail prevents us from being able to examine factors that have been identified as being important in other studies of outcomes at younger ages e.g. placement stability, length of time in care, reason for being in care. We are therefore unable to speculate on how these factors would impact on our later outcomes. We have expanded the part of the discussion which mentions this limitation and added a reference:

"The ALSPAC measures on looked-after status and adoption lack detail. The age a child enters care, and the number of placements they have, are thought to be key factors in determining the likelihood of positive outcomes (REF), but we do not have this information on our participants. We do not know the reason for their care status, or the age they left the care system. Some of the adoptions may have been by step-parents, rather than as a result of being removed from birth parents. Some of the adopted individuals had also been in care, but we do not know why or for how long."

3. Additionally, another comment that is not possible to address, but your substance use questions seem basic - only address cigarette smoking, alcohol and cannabis - what about other illegal drugs?

Response: Tobacco and alcohol are the two most commonly used substances, and cannabis the most commonly used illegal substance, in the UK. All three can have significant detrimental effects on health; therefore understanding determinants of their use and identifying high risk groups is of public health importance. We would have liked to also include other illegal substances in our study but the number of participants who reported taking them was too small. However, we do include a measure of 'ever having had an addiction', and it is likely that some of these addictions will have been to other illegal substances. We have added the following to this section of the Discussion text:

“The numbers reporting use of other illegal substances was too small to analyse, however the ‘ever had an addiction’ measure is likely to include addictions to other illegal substances.”

4. How did the study deal with children who were adopted from care? Were they grouped with the adoptees or the looked-after group? This needs to be addressed in description of methods/participants. It should also be addressed in discussion as a potential confound/overlap in sample.

Response: In the Methods, we describe our exposure measure - individuals who were adopted were not included in the looked-after group:

“A 3-category exposure variable was derived defining childhood care status: not looked-after or adopted; looked-after (not adopted); adopted.”

Further, in the second paragraph of the results we describe how many of the adopted group reported they had also been in care:

“A minority of the adoptees had also been in care (women 14%, men 16%), with a similar proportion reporting that they didn’t know.”

Therefore a substantial minority of the adopted group don’t know if they have been in care or not – perhaps reflecting that they were adopted at a very young age, and/or adopted by their foster carers thereby blurring the distinction between being in care and adopted. We have no information on whether any of the adoptees were formally ‘adopted from care’.

In the discussion, we have expanded our description of the limitation of our adoption measure:

“Some of the adoptions may have been by step-parents, rather than as a result of being removed from birth parents. Some of the adopted individuals had also been in care, but we do not know why or for how long.”

5. A few times in the paper, "adjustment" is mentioned. I would assume the authors mean "adjustment for SEP" but this is unclear. This issue arises on page 9 three times.

Response: In the analyses section of the methods we detail which variables are adjusted for – these are mainly, but not exclusively, related to SEP:

“Models were run unadjusted and adjusted for age and measures related to SEP (relationship status, education, financial difficulties, social class and housing tenure). Models also adjusted for parity for women, and for whether the pregnancy was intentional for the pregnancy time-point outcomes, and for pregnancy intentions for the 5-year time-point outcomes.”

The footnotes of Tables 3-5, and supplementary Tables G-I, also list these variables.

To add clarity, we have added the following sentence to the Analyses section of the methods:

“Where results are described as being ‘adjusted’ in the results text, this means after adjustment for age, all of the SEP variables, plus parity and pregnancy-intention variables where relevant.”

6. The reporting of your results is very basic, and the tables lack statistical information. Where are the logistic regression coefficients? Standard errors? Wald statistics and degrees of freedom? I see this as a crucial omission and would not accept any paper without these data.

Response: We are aware that in psychology and social sciences, logistic regression results are often presented in the APA style, which we believe is what the reviewer is suggesting. We believe that our presentation of results is in the format most commonly used in epidemiology, and also in BMJ Open. In the paragraphs below we summarise the results that we have presented.

The logistic regression coefficients (odds ratios) and 95% confidence intervals are given for the unadjusted and adjusted associations for the following: care status and substance use for men and women at both time points (Table 3); care status and mental health for men and women at both time points (Table 4); care status and criminal involvement and social support for men and women (Table 5). The supplementary tables G-I present odds ratios and 95% CI for associations between care status and substance use, mental health, and social support respectively for men and women in the cross sectional sample.

Tables 1 and 2 provide summary statistics for the women and men respectively. No regression coefficients are included in these tables as they are not the results of regression analyses. Instead we present percentages with 95% CI, with the exception of age which is summarised as a mean and 95% CI. Similarly, in the supplementary Tables, Table B summarises childhood/adolescent measures by presenting percentages and 95% CI; and Tables C and D summarise the women and men's cross-sectional sample respectively, again using percentages, means, and confidence intervals.

In all these Tables, we include confidence intervals as these give the range of values in which we are 95% confident the true population parameter lies. We believe this is more informative for the reader than simply stating standard errors or p-values. While these could also be added to the Tables we believe they would clutter them unnecessarily, making them less reader-friendly.

There are some figures in the paper that are not presented with confidence intervals. Firstly, supplementary Table A (summary of missing data) and secondly, some sample descriptives related to experiences of care (e.g. type of care setting, length of time in children's home) in the first section of the results.

We do not feel it is necessary to state confidence intervals around these percentages as these variables, which we hope will be of interest to readers, have limitations (as mentioned by Reviewer 2, Comment 2) which mean they are not included in the formal analyses. Further, while we present descriptives for the care experiences of men and women separately, we did not undertake a formal statistical comparison between the two as to do so would be inappropriate given that the samples are not independent (as the men are partners of the women).

7. In the discussion section - does the BCS70 report on only cannabis use or does it include other illegal drugs - this needs to be more clearly specified for readers who are not intimately familiar with BCS70. Additionally, the timing of the BCS70 needs to be addressed with respect to your comparisons. Is it really that there is low drug use in your sample, or are there chronosystem differences in population drug use?

Response: The BCS70 paper measure related to the use of all illegal substances. We have now made this clearer in our paper:

“The BCS70 study included a measure of ‘illegal substance use in past year’, without further breakdown, and found high levels in the men; 26% of those never in care and 34% of those who had been in care (REF).”

The BCS70 study participants were born in 1970 and outcomes reported at age 30, the ALSPAC men and women were born mainly in the 1960s and reported outcomes in their late 20s and 30s. Some of the difference could possibly be attributed to generational differences and a cohort effect. But we think the most likely explanation is that the ALSPAC adults were the ones recruited into the study, so there was a selection effect on them (i.e. as discussed in paper, they had to be expecting a baby) whereas it was the parents of the BCS70 sample who were recruited to that cohort study – the adult sample are the babies who were born in 1970 and have been studied ever since. We have added this point to our paper:

“Part of the discrepancy in drug use between the two studies could reflect differences in sample selection: the ALSPAC sample were recruited to the study as adults who were expecting a baby, whereas the BCS70 sample were in that study from birth, as it was their parents who were recruited.”

8. The discussion section needs much more analysis, interpretation and conclusion. This and the statistical omissions are the greatest issues with this manuscript. What do your results mean, and what should we do with this information?

Response: Thank you for highlighting this important point. We have revised our ‘Implications for practice and policy’ section to make it much more focused on what our results mean, who they are most relevant for, and also what needs to happen next. These changes were detailed in response to reviewer 1 above. Given the limitations in our observational data that we acknowledge, and that both reviewers correctly highlight, we feel it is important not to over interpret the meaning of our results.

9. On page 11, the paragraph on mental health should likely include a comment about adoptive family follow-up. Research suggests that adoptees and adoptive families may be more comfortable accessing services due to the intimate service use during and immediately following the adoption process. It could be that adopted persons are more comfortable using mental health services that would prevent any mental health issue from escalating to your high thresholds.

Response: Thank you for this comment. The reviewer is correct in suggesting that adoptees may be more comfortable accessing mental health services than their general population peers. However, it was not possible to test if this assumption is true for this sample of adoptees as data on accessing mental health services was not available within the dataset. Therefore, we have not added these points to our paper

10. Your tables could use better delineation of the sample sizes in cells/rows.

Response: We state percentages rather than sample sizes in Tables 1-5 as these results are from imputed data, therefore the n differs across the imputed datasets. The percentages shown are from results aggregated across all the imputed datasets using Rubin’s rules. We have added footnotes to Tables 1-5 to explain this. In Supplementary Tables E and F the n is given for each column as these are results from complete case analyses.

11. Finally, given that you have a large sample size and binary indicators, it would have been very interesting to see a latent class analysis with these data. The logistic regression is basic and your sample could lend itself to much more complicated and rich analyses.

Response: We took a parsimonious approach to our analysis using the simplest methods appropriate to our data and our aims. We wished to examine associations for each of our outcomes individually, to determine which outcomes were associated with care status and to see if there was a consistent pattern or otherwise across the care status categories. These details would have been lost had we aggregated outcomes using latent class analysis. Further, keeping the outcomes separate aided comparison with previous literature. We feel that the techniques we use are appropriate to meet the aims of this paper and that a more elaborate approach, such as latent class analysis, would not have been more informative in this context. However beyond this paper, we agree with the reviewer that further research on this sample could explore some of the associations found in more detail using other statistical approaches.

VERSION 2 – REVIEW

REVIEWER	Jessica A.K. Matthews University of Massachusetts-Amherst, Amherst, Massachusetts, United States
REVIEW RETURNED	11-Dec-2017
GENERAL COMMENTS	It seems the authors have thoughtfully and thoroughly addressed comments and concerns from both reviewers. If the presentation of statistical results is in line with the field of epidemiology and the journal, I recommend the manuscript is accepted.